# Activation/Inhibition of Gene Expression Caused by Alcohols: Relationship with the Viscoelastic Property of a DNA Molecule

**DOI:** 10.3390/polym15010149

**Published:** 2022-12-28

**Authors:** Kohei Fujino, Takashi Nishio, Keita Fujioka, Yuko Yoshikawa, Takahiro Kenmotsu, Kenichi Yoshikawa

**Affiliations:** 1Faculty of Life and Medical Sciences, Doshisha University, Kyoto 610-0394, Japan; 2Cluster of Excellence Physics of Life, Technical University of Dresden, 01307 Dresden, Germany

**Keywords:** transcription/translation efficiency, ethanol, propanol isomers, single DNA observation, intramolecular fluctuation, spring constant, damping constant

## Abstract

Alcohols are used in the life sciences because they can condense and precipitate DNA. Alcohol consumption has been linked to many diseases and can alter genetic activity. In the present report, we carried out experiments to make clear how alcohols affect the efficiency of transcription-translation (TX-TL) and translation (TL) by adapting cell-free gene expression systems with plasmid DNA and RNA templates, respectively. In addition, we quantitatively analyzed intrachain fluctuations of single giant DNA molecules based on the fluctuation-dissipation theorem to gain insight into how alcohols affect the dynamical property of a DNA molecule. Ethanol (2–3%) increased gene expression levels four to five times higher than the control in the TX-TL reaction. A similar level of enhancement was observed with 2-propanol, in contrast to the inhibitory effect of 1-propanol. Similar alcohol effects were observed for the TL reaction. Intrachain fluctuation analysis through single DNA observation showed that 1-propanol markedly increased both the spring and damping constants of single DNA in contrast to the weak effects observed with ethanol, whereas 2-propanol exhibits an intermediate effect. This study indicates that the activation/inhibition effects of alcohol isomers on gene expression correlate with the changes in the viscoelastic mechanical properties of DNA molecules.

## 1. Introduction

Alcohol precipitation is widely adapted to purify or concentrate nucleic acids in life sciences. Ethanol and 2-propanol are frequently used to isolate DNA molecules from cells through precipitation, whereas 1-propanol is not. The effects of low molecular-weight alcohols on DNA precipitation have been discussed in relation to the phenomenon of DNA condensation [1,2]. Fluorescence microscopy recently showed that DNA exhibits a reentrant transition above 50% ethanol (i.e., elongated coil to compact state, and then into elongated coil) when a single DNA molecule was observed with increasing concentrations of ethanol [3]. Secondary DNA structure exhibits a successive change, B → C → A, when measured with a CD (circular dichroism) spectra [2,4,5,6]. A comparative study between single DNA observation and CD measurement indicated that the reentrant transition of the secondary structure corresponds to the reentrant transition of the higher-order structure (i.e., coil → compact → coil conformation) [3]. A similar reentrant transition of the higher-order structure of DNA was also observed with increasing concentrations of 1-propanol above 50% [7]. However, individual DNA molecules underwent a continuous transition, with gradual shrinking, from an elongated coil into a compact globule with 2-propanol.

Ethanol and 2-propanol exhibit germicidal effects against viruses, bacteria, and fungi, and are currently used as disinfectants. As for the biological effects of ethanol as a drink, excessive alcohol consumption leads to prevalent liver diseases, AUD (alcohol use disorder), and other various diseases [8]. Alcohol consumption, especially through adolescent binge drinking, causes epigenetic reprogramming on the synaptic activity response element [9]. Based on a recent statistical analysis of human participants (ca. 4.7·10^5^), alcohol intake above threshold levels was shown to shorten the length of telomeres, which is a causal risk factor for several age-related diseases such Alzheimer’s disease [10]. Many studies have been performed to identify specific genes whose expression is modified after alcohol exposure in animals and cultured cells [11,12,13,14]. Genome-wide association studies (GWAS) reported that alcohol consumption caused both the down- and up-regulation of hundreds of genes in mouse experiments. Importantly, these studies showed that gene expression patterns correlated well between mice and humans. They also discussed the long-standing genomic effects (e.g., genetic and epigenetic modifications) caused by alcohol consumption. As far as we know, the current knowledge on the effects of relatively low concentrations of low molecular-weight alcohols on genetic activity is rather poor, despite their wide usage and application in biological sciences and also in daily life as mentioned above. Thus, in the present study, we carried out an experimental study to compare the effects of low concentrations of alcohols (i.e., ethanol, 1-propanol, and 2-propanol) on gene expression, in addition to the single molecular observation on the influence of different alcohols on the dynamic property of the higher-order structure of DNA through the analysis of intrachain Brownian motion.

To discern alcohol’s effects on DNA’s biological functions, we have performed gene expression experiments using cell-free expression systems with DNA (TX-TL) or mRNA (TL) templates. In the present study, we show the effects low concentrations of alcohols have on gene expression with TX-TL and TL reaction mixtures. The TL reaction was accelerated by the addition of 2–4% of alcohol (ethanol, 1-propanol or 2-propanol), but the promotion effect of 1-propanol was weaker than the others. The addition of 2–4% ethanol or 2-propanol to the TX-TL reaction increased gene expression, while 1-propanol tended to depress expression at similar concentrations. At concentrations higher than 7%, ethanol, 1-propanol and 2-propanol all inhibited the TX-TL and TL reactions. To gain insight into the underlying mechanism for such a significant effect on genetic activity by relatively low concentrations of alcohols, we investigated how alcohols might affect the conformational properties of DNA molecules. Current studies have reported marked effects on the conformation of DNA, but only for alcohol concentrations above 50% [2,4,5,15,16,17]. To determine the effects of low alcohol concentrations on DNA conformation, we analyzed the time-dependent change of intramolecular Brownian fluctuation of single DNA molecules [18] and found a large effect on the viscoelastic property of single DNA molecules. Interestingly, changes in the viscoelastic property of DNA, caused by low concentrations of alcohols, provide useful insights into the mechanism of gene expression promotion/inhibition observed in the present study.

## 2. Materials and Methods

### 2.1. Materials

Ethanol, 1-propanol, 2-propanol, and 2-mercaptoethanol (2-ME; an antioxidant) were purchased from Wako Pure Chemical Co., Ltd. (Osaka, Japan). T4 GT7 bacteriophage DNA (166 kbp, contour length 57 μm) was purchased from Nippon Gene Co., Ltd. (Tokyo, Japan). The fluorescent cyanine dye, YOYO-1 (1,10-(4,4,8,8-tetramethyl-4,8-diazaundecamethylene(bis(4-((3-methylbenzo1,3-oxazol-2-yl) methylidene)-l,4-dihydroquinolinium) tetraiodide) was purchased from Molecular Probes Inc. (Eugene, OR, USA). Plasmid DNA (Luciferase T7 Control DNA, 4331 bp) containing a firefly luciferase gene and a T7 RNA polymerase promoter sequence was purchased from Promega (Madison, WI, USA). The mRNA (Luciferase Control RNA), which is uncapped in vitro-transcribed RNA containing a 30-base poly(A) tail, was used to produce luciferase and was purchased from Promega (Madison, WI, USA). Other chemicals were obtained from commercial sources and were analytical grade.

### 2.2. Luciferase Assay for Gene Expression (TX-TL)

A cell-free in vitro luciferase assay was performed with a TnT (Rabbit Reticulocyte Lysate) T7 Quick Coupled Transcription/Translation System (Promega) according to the manufacturer’s instructions (www.promega.com/protocols/, accessed on 20 October 2022) and as previously described by our group [19,20,21,22]. Plasmid DNA (Luciferase T7 Control DNA, 4331 bp, Promega, Madison, WI, USA) encoding the firefly luciferase gene was used as the DNA template. The DNA concentration was 0.3 μM in nucleotide units. The reaction mixture containing the DNA template was incubated for 90 min at 30 ℃ on a Dry Thermo Unit (TAITEC, Saitama, Japan). The incubation time (90 min) was optimized so that the luminescence intensity was in the linear phase and had not yet reached the rate-slowing stage, as shown in Appendix A. We measured the change in the luminescence intensity when various concentrations of ethanol, 1-propanol, or 2-propanol were added to the reaction mixture. The expression of luciferase was evaluated after the addition of luciferin (luciferase substrate; Luciferase Assay Reagent, Promega) by detecting the emission intensity at 565 nm using a luminometer (MICROTEC Co., Chiba, Japan).

### 2.3. Luciferase Assay for translation (TL)

Uncapped RNA containing a 30-base poly(A) tail (Luciferase Control RNA, Promega) was used as the RNA template. The RNA concentration was 0.3 μM in nucleotide units. The reaction mixture was incubated for 60 min under similar conditions as in the TX-TL reaction. The incubation time was optimized and measured as described for the TX-TL reaction.

### 2.4. Fluorescence Microscopy (FM) Observation

DNA was dissolved in a 10 mM Tris-HCl buffer solution at pH 7.5 with 4% (*v*/*v*) 2-ME, and a final DNA concentration of 0.1 μM. Measurements were conducted with 0.05 μM of the fluorescence dye, YOYO-1 (Excitation/Emission: 491/509 nm), as we have previously described [20,21,22]. Single-molecule observation on DNA was performed with an inverted fluorescence microscope (Axiovert 200, Carl Zeiss, Oberkochen, Germany) equipped with a 100× oil-immersion objective lens. Fluorescent illumination was provided by a mercury lamp (100 W) through a filter set (Zeiss-10, excitation BP450–490; beam splitter FT 510; emission BP 515–565). The observed images were recorded on a DVD at 30 frames per second through the high-sensitivity EBCCD (Electron Bombarded Charge-Coupled Device) camera (Hamamatsu Photonics, Shizuoka, Japan) and analyzed with the image-processing software, ImageJ (National Institute of Mental Health, MD, USA). We evaluated the time-dependent change of the long-axis length of DNA by analyzing the fluorescence images on each video frame.

## 3. Results

### 3.1. Effects of Alcohol Concentration on the Efficiency of Gene Transcription/Translation

We examined the effects of alcohols on the activity of gene expression (transcription-translation; TX-TL) using a cell-free in vitro luciferase assay with the TnT (Rabbit Reticulocyte Lysate) T7 Quick Coupled Transcription/Translation System. Protein synthesis activity measured as gene expression (TX-TL) at various concentrations of ethanol, 1-propanol and 2-propanol (1–10%) is shown in Figure 1a as the relative luminescence intensity of the luciferin-luciferase reaction (The detailed data together with additional explanation are shown in Appendix A). The vertical axis (the relative luminescence intensity) was normalized to the control experiment observed in the absence of any alcohol. Around 2–3% of ethanol or 2-propanol promoted TX-TL expression, whereas 1-propanol depressed it. Above 7%, all three alcohols almost completely inhibited the TX-TL reactions. Figure 1b shows the effect of alcohols on translation, TL, as measured by a change in the relative luminescence intensity. Interestingly, below 2% all three alcohols promoted TL reaction. On the other hand, for the TX-TL reaction, 2-propanol exhibited an inhibitory effect in contrast to the activation by ethanol and 1-propanol for TX-TL. It is noted that such unique effects of alcohols would be apparent even below 1% as expected from the graph in Figure 1, which may be associated with realistic concentrations in certain tissues for heavy drinkers. The promotion efficiencies of all the alcohols tended to decrease above 4% and near zero at 10%. Thus, these alcohols cause a bimodal effect (i.e., a promotion at lower concentrations and inhibition at higher concentrations) for both TX-TL and TL reactions, with the exception of a continuous inhibitory effect caused by 1-propanol for TX-TL.

### 3.2. Effect of Alcohols on the Viscoelasticity of a Single DNA Molecule Evaluated Using Thermal Fluctuations

As indicated above, the efficiencies of transcription-translation (TX-TL) and translation (TL) were markedly changed at relatively low concentrations of alcohol. To clarify how alcohols caused these effects, we measured the conformational dynamics of individual T4 GT7 DNA molecules using fluorescence microscopy. Examples of time-dependent fluctuations in the DNA conformation are shown in Figure 2, where fluctuations of the DNA conformation were observable due to the relatively large size of the DNA molecule (166 kpb with a contour length of 57 µm). By calculating the intramolecular Brownian motion of individual DNA molecules, we noticed that the degree of fluctuation tended to be depressed with the addition of alcohols. In other words, DNA molecules appear more constrained with an increase in alcohol concentration. Therefore, we analyzed time-dependent fluctuations by measuring the long-axis length, *R*, by adapting the similar procedure of our previous report [18]. Based on the data of the time-dependent change of the long-axis length exemplified in Figure 3, we have evaluated the autocorrelation *C(τ)* of the long-axis length.
(1)$$C\left(\tau \right)=\langle R\left(\tau \right)-\bar{R}\rangle \langle R\left(0\right)-\bar{R}\rangle \;$$

$\bar{R}$ is the time-average of the long axis-length, *τ* is the time difference between two data-points, and the symbol, < >, means the average of the time-dependent variable. Figure 4 shows the autocorrelation functions calculated from the time-dependent fluctuation of the long-axis length.

Based on the framework of fluctuation-dissipation theory for the time-dependent characteristic of harmonic spring under thermal (Brownian) fluctuation, the autocorrelation function is represented in Equation (2) [23,24].
(2)$$C\left(\tau \right)\sim \;\frac{k_{B}T}{k}e^{-\gamma \tau }\cos\omega \tau$$
where *k_B_* is the Boltzmann constant, *T* is the absolute temperature, *k* [N/m] is the spring constant, *γ* [sec^−1^] is the damping constant, and *ω* is the angular frequency. The spring constant *k* is calculated from the relationship of $k\;\approx \;\frac{k_{B}T}{C\left(0\right)}$ from the initial value of the autocorrelation function, *C*(0) at *τ* = 0. Fitted curves, based on Equation (2), to the experimental results, were used to determine the spring constant *k*, damping constant *γ*, and angular frequency *ω*. These evaluated parameters are given in Appendix A. The almost linear relationship between *k* and *ω^2^*, as observed in Appendix A, suggested that the changes in the spring constants estimated from *C*(0) correspond well to those based on *ω* (i.e., evaluated from different parts of the fitting curve).

Figure 5 shows changes in the spring constant (*k*) and the damping constant (*γ*) at different alcohol concentrations as evaluated through the procedure explained above. Both *k* and *γ* values almost double at 2% alcohol, and the differences among the alcohols were not significant. At 4%, 1-propanol increased both *k* and *γ*. 4% 2-propanol also increased *k*, but not to the same level as 4% 1-propanol. In relation to the specific effect on the viscoelastic property induced by 1-propanol compared to ethanol and 2-propanol, 1-propanol caused greater effects in the TX-TL and TL reactions compared to ethanol and 2-propanol (see Figure 1).

## 4. Discussion

In the present study, we clarified the effects that relatively low concentrations of alcohols have on DNA expression (i.e., transcription-translation and translation) using TX-TL and TL reactions, respectively. We also investigated the dynamic conformational property of genome-sized DNA. Specifically, we focused on the effects low concentrations of alcohols (i.e., ethanol, 1-propanol, and 2-propanol) have on some biological functions and DNA conformation. Strong effects on the bio-function of genetic activity and DNA conformation were observed for these alcohols.

First, we will discuss the promotion/inhibition caused by low concentrations of alcohol (Figure 1). During the past several decades, cell-free gene expression systems have been developed as a useful in vitro model to gain insight into the underlying mechanism of transcription–translation (TX-TL) in living cells and have been actively applied to various subjects in the biological and medical sciences. In addition to key-lock type specific interactions, environmental parameters in the cytoplasm have a substantial effect on gene expression activity. For example, macromolecular crowding causes either the promotion or repression of both TX and TL in vitro experiments using cell lysates [25,26,27]. Using these cell-free gene expression systems, we found that polyamine, with a trivalent positive charge (3+), completely inhibits protein synthesis above a certain threshold concentration [21]. Below this critical concentration of 3+ ions, gene expression activity is normally much higher in the cell lysate. In other words, polyamines exhibit a bimodal effect (e.g., promotion/inhibition) on the activity of gene expression. Interestingly, monovalent cations, such as Na^+^ and K^+^, also show the bimodal effect of promotion/inhibition on gene expression, which was interpreted as the effect of the competition of monovalent cations with multivalent polyamines [28]. In the present study, we have extended these earlier studies on how environmental parameters (not the key-lock interaction) affect the genomic activity of DNA. Using low concentrations of alcohols in TX-TL and TL reactions, we found a bimodal effect on the genetic activity caused by ethanol and 2-propanol, promotion below 4% and inhibition above 7% (Figure 1). In contrast, 1-propanol has almost no promotion and inhibits both TX-TL and TL reactions. Interestingly, this type of bimodal effect (i.e., promotion/inhibition) with ethanol and 2-propanol has been found for the environmental (not the key-lock interaction) parameters such as the concentrations of polyamine, as well as monovalent and divalent cations.

Next, we will discuss the effect low alcohol concentrations have on the manner of intrachain fluctuation (Figure 2, Figure 3, Figure 4 and Figure 5). During the past several decades, many studies have observed higher-order structural changes on relatively large DNA molecules above several tens kbp. For example, electron microscopy and atomic force (scanning probe) microscopy have provided useful information on the conformation or higher-order structure of DNA molecules [29,30,31,32]. Although these methodologies generate high-resolution images of DNA on the order of nm to sub-nm, these observations are performed for DNA molecules on a solid surface. In other words, it is difficult to gain information on native DNA molecules without the interference effect from the solid surface. Recent developments in cryo-electron microscopy allow the imaging of biomacromolecules, including DNA molecules, without the effect of a solid surface [33,34]. Unfortunately, this methodology provides 3D images by excluding the time-dependent fluctuation information. Single-molecule force spectroscopy is a useful methodology that measures the elasticity of DNA molecules by applying optical or magnetic tweezers [35,36,37,38,39]. Despite the usefulness of such methodologies to measure the elasticity of a single DNA molecule, reliable data can be obtained only for a region with relatively large stretching. In other words, it is difficult to gain information on the elasticity without the stretching (i.e., the potential profile near the free energy minimum in the absence of external stimuli, such as the extension by the tweezers). A theoretical consideration for the elasticity of a single semi-flexible polymer suggests that its spring constant *k* can be simply evaluated as in the following relationship [18,37,40,41,42,43]:$$k\;\sim \;k_{B}T/l_{p}L_{c}$$
where *l_p_* and *L*_c_ are the persistence length and contour length, respectively. For the T4 GT7 DNA used in the present study, we can estimate *l_p_* ≈ 50 nm and *L*_c_ ≈ 57 μm. By considering that *k_B_T* ≈ 4.1 pN·nm at room temperature, we can estimate the spring constant *k* of T4 GT7 DNA to be about 1.4 nN·m^−1^ (~10^−9^ N·m^−1^) based on the above-mentioned simple theoretical model for a semiflexible polymer. On the other hand, from the single molecular observation on the fluctuation, the spring constant *k* of the control (in the absence of alcohol) is found to be 12 nN·m^−1^.

Essentially the same spring constant value of T4 GT7 DNA in an aqueous solution was obtained in our past study, which was measured through a similar procedure as in the present study [18] and used here as the control. Although this value is several times larger than the above-mentioned simple theoretical assumption, the order of the elasticity strength would be rather similar. In the above theoretical framework, only the contribution of bending stiffness, or persistence length, is used to evaluate the elasticity of a single semiflexible polymer chain. For actual DNA molecules, other contributions such as twisting and interactions between different segments will contribute to the apparent elastic property. In addition to the observed mechanical characteristics of single DNA, other freedoms of time-dependent intramolecular fluctuation besides the motions of the long-axis (e.g., rotational fluctuations) contribute to the apparent viscoelasticity [44].

Alcohols affected the viscoelastic properties of DNA (Figure 5) and an increase in both the elasticity and viscosity was observed in the following order: 1-propanol > 2-propanol > ethanol. This ordering follows the same order of inhibitory effects on TX-TL and TL reactions (Figure 1). Thus, it is plausible that the ability of alcohols to increase the stiffness of the higher-order DNA structure directly correlates with its gene expression activity. Hereafter, it may be of interest to measure how alcohols affect the viscoelasticity of mRNA. The promotion of TX-TL and TL reactions by alcohols exhibits the opposite ordering: ethanol > 2-propanol > 1-propanol. Although it is difficult to ascertain the underlying mechanism of the promotive effect at present, a weak increase in the viscoelasticity of DNA may correlate with transcription and translation efficiency. We recently reported the effect of 1-propanol and 2-propanol on the conformation of DNA based on the single DNA observation and CD measurement [7]. This study found that the hydrodynamic radius *R*_H_ of T4 GT7 DNA tends to decrease when alcohol is added. The approximate *R*_H_ is 0.9 µm in an aqueous solution without alcohol but goes down to 0.4 and 0.5 µm when 1-propanol (10%) and 2-propanol (10%) are added, respectively. Thus, the shrinking effect on DNA is larger with 1-propanol than with 2-propanol, which is attributable to the larger hydrophobic property of 1-propanol with a straight-chained chemical structure.

We have shown that it is possible to evaluate both the elasticity and viscosity (damping constant) in an aqueous environment from the analysis of the autocorrelation function on the fluctuation of single DNA molecules in the present study. It becomes clear that long DNA, such as T4 GT7 DNA, is characterized by the mechanical properties of an underdamping object.

Such characteristics are expected to contribute to the biological activity of genomic DNA in a living cellular environment. Further studies on the relationship of the physicomechanical property of genome-sized DNA with genomic activities would provide a basic understanding of biosciences. The apparent viscoelastic property observed in the present study could be explained by incorporating the effect of nonlinear cooperation between bending and twisting together with the actual potential between DNA segments [45,46,47,48]. We now discuss the structure-activity relationship in the presence of low concentrations of alcohols based on the above arguments for how alcohols affect TX-TL and TL reactions and the mechanical property of DNA. Figure 1 shows a large difference in the manner of promotion/inhibition caused by 1-propanol compared to ethanol and 2-propanol both for TX-TL and TL reactions. 1-propanol (2%) exhibits an inhibitory effect on TX-TL in contrast to its promotion effect on TL. This implies that the large difference between 1-propanol and ethanol or 2-propanol is mostly due to the inhibitory effect on transcription (TX). As for the effect of alcohols on the viscoelasticity of DNA (see Figure 5), the spring and damping constants are rather similar among alcohols at 2%. A difference among these alcohols becomes apparent at 4%, where the effect of 1-propanol is much larger than the other alcohols. It is noted that, in the present study, the solution conditions between in vitro gene expression and viscoelasticity (thermal fluctuation measurements) experiments were different, i.e., the former adapted cell-free extracts and the latter used Tris-HCl buffer solution. Regardless of the difference in solution conditions, it may be quite plausible that the larger effect of 1-propanol on the mechanical property of DNA may be directly related to its marked inhibitory effect on transcription under a rather low alcohol concentration of around 2%.

Various environmental/nonspecific parameters, without key-lock type characteristics, can significantly influence genetic activity (e.g., the manner of confinement with the phospholipid membrane, concentrations of monovalent or divalent cations, polyamines, cationic oligopeptides, noncoding RNAs, NTPs (ribonucleoside triphosphates), and the size of template DNA, etc.) [20,28,49,50,51,52,53,54,55,56,57]. As for the influence of these environmental parameters, competitive/cooperative interactions of negatively charged phosphate groups in DNA with various cations species in solution will play important roles. According to the counterion condensation theory, around 76% of the intrinsic negative charges of phosphate groups are neutralized in physiological solutions by attracting monovalent cations from the environment in the absence of multivalent cationic species [58,59]. That is, only 24% remain dissociated to provide a negative charge on DNA [60]. The counterion condensation theory interprets the large different effect of counter ions from the current theoretical hypothesis of the Debye–Hückel theory, owing to the strong correlation effect of negatively charged residues aligned along the double-stranded DNA [61]. In the Debye–Hückel framework, the shielding efficiency of electronically charged species is interpreted as the sum of the effect of the surrounding counterions with different valences; Debye length $\lambda_{D}\;\sim \;I^{-1/2}$, where ionic strength $I\;\sim \;\sum c_{i}Z_{i}^{2}$. On the contrary, the degree of counterion condensation is not simply interpreted in terms of the ionic strength *I*, but instead through a competitive effect among different countercations [62]. Although counterion condensation does not interpret the behavior of charged species for highly charged rod-like polymers in a satisfactory manner, the importance of the correlation effects owed to the long-range Coulombic interaction is still regarded as relevant [63,64,65]. An increase in the degree of counterion condensation, when alcohols are added to the aqueous solution, may be expected because of their influence to decrease the dielectric constant of a solution. Among the alcohols examined in the present study, it is plausible that the longer alkyl chains on 1-propanol (compared to those in ethanol and 2-propanol) localize more readily to relatively hydrophobic regions on double-stranded DNA. This specific characteristic of 1-propanol may explain the increase in elastic and damping constants and why there is a greater depression effect on gene expression. On the other hand, the promotion of gene expression by low concentrations of ethanol and 2-propanol is attributed to relatively complex effects including the physicochemical influences on RNA polymerase and substrates for the transcriptional reaction, in addition to the ionic environment of DNA.

Quite recently, Krishnan et al. reported that rats acutely exposed to ethanol (oral) had epigenetic and transcriptomic changes by using next-generation sequencing methodologies, ATAC-seq (assay for transposase-accessible chromatin with high throughput sequencing), and RNA-seq (RNA sequencing) [66]. They concluded that acute ethanol exposure can ‘open’ chromatin structure, resulting in changes to DNA-protein interactions and the transcriptome. Further studies that combine in vitro measurements and animal experiments are expected to unveil the underlying mechanism/s of low alcohol concentrations on genetic and physiological activity.

## 5. Conclusions

Our main conclusions, together with future perspectives, on the effects of alcohols (ethanol, 1-propanol, and 2-propanol) on gene expression and the conformational property of DNA are as follows:

Effect of alcohols on gene expression. We examined transcription and translation efficiencies by adapting cell-free gene expression systems with plasmid DNA (TX-TL) and mRNA (TL) templates, respectively. These alcohols cause a bimodal effect (i.e., a promotion at lower concentrations (around 2–3%) and inhibition at higher concentrations) for both TX-TL and TL reactions, except for the constant inhibitory effect of 1-propanol on TX-TL.

The effect of alcohols on the mechanical property of DNA. Time-dependent conformational fluctuation analysis of T4 GT7 DNA (166 kbp), with fluorescence microscopy, shows that 1-propanol tends to markedly increase both spring and damping constants of single molecule DNA, in contrast to the weak effects observed with ethanol and 2-propanol.

1-propanol appears to favor relatively hydrophobic localization on double-stranded DNA, because of its longer alkyl chain compared to those in ethanol or 2-propanol. This specific characteristic of 1-propanol may explain the increase of elastic and damping constants and why gene expression is depressed. On the other hand, the promotion of the gene expression caused by low concentrations of ethanol and 2-propanol could be attributed to relatively complex factors, including the ionic environment of DNA, the physicochemical influences on RNA polymerase, and the substrates for the transcriptional reaction. Further studies combining in vitro measurements and animal experiments are expected to unveil the underlying mechanism of how low concentrations of alcohols alter genetic and physiological activities.

We have adopted useful experimental methodologies in the present article including (i) measurements of cell-free gene expression using plasmid DNA (TX-TL) and mRNA (TL) templates and (ii) evaluation of the spring and damping constants of single DNA molecules in solution without any external stress. Further application of these methodologies to uncover the various problems in biological and medical sciences is highly expected.

## Figures and Tables

**Figure 1 polymers-15-00149-f001:**
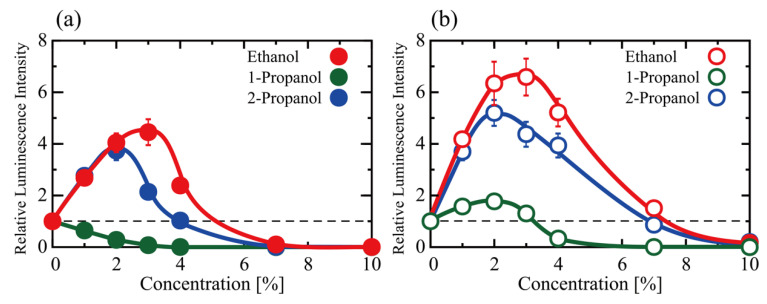
The effect of ethanol, 1-propanol, and 2-propanol on gene expression, (**a**) transcription-translation (TX-TL) and (**b**) translation (TL). The vertical axis shows the relative luminescence intensity of the luciferin-luciferase reaction, which corresponds to the efficiency of luciferase production. Luminescence was measured after 90 min of incubation at 30 °C. The intensity was normalized to the control (i.e., no alcohol in the reaction mixture). The concentration of DNA (Luciferase T7 Control DNA) in (**a**) and the concentration of mRNA (Luciferase Control RNA) in (**b**) was 0.3 μM. Data presented as mean ± SEM.

**Figure 2 polymers-15-00149-f002:**
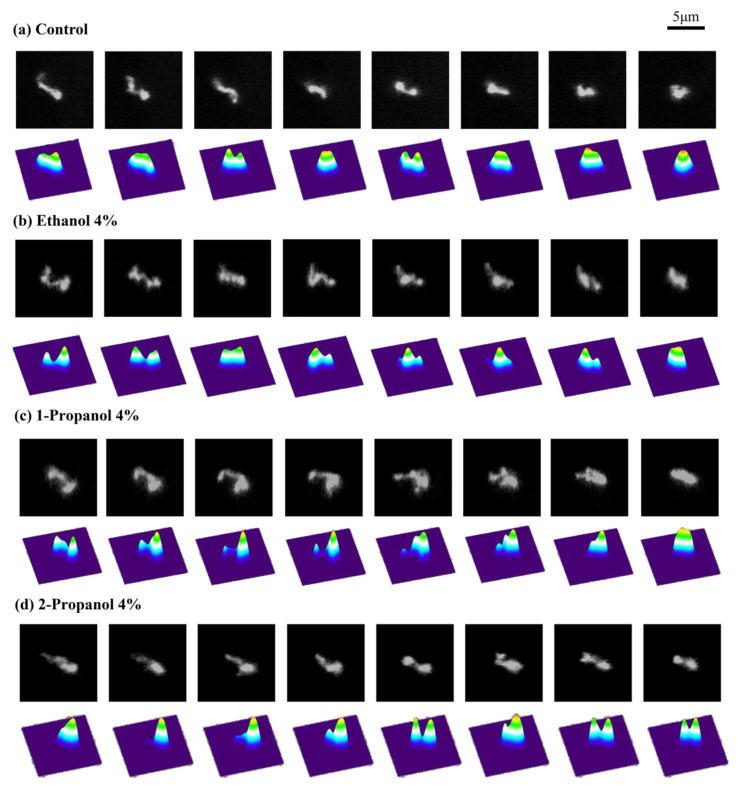
Representative time-dependent fluctuation of single T4 GT7 DNA molecules under Brownian motion observed by fluorescence microscopy (FM). The time interval between neighboring frames is 0.1 s. The pseudo-3D profiles on the lower frames of each image indicate the fluorescence intensity, corresponding to the spatial density of the segments in a single DNA. (**a**) Aqueous solution of 10 mM Tris-HCl buffer solution without alcohol (control), (**b**) with 4% ethanol, (**c**) with 4% 1-propanol, and (**d**) with 4% 2-propanol.

**Figure 3 polymers-15-00149-f003:**
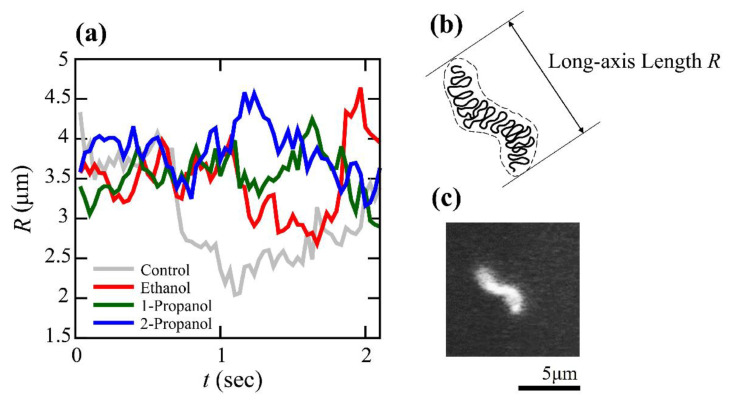
Time-dependent change in the long-axis length, *R*, of T4 GT7 DNA molecules as a measure of intrachain Brownian motion. (**a**) Time traces of the long-axis length at different solution conditions. Ggray line (control) is the measurement obtained using 10 mM Tris-HCl buffer solution at pH 7.5 with 4% (*v*/*v*) 2-ME without the addition of alcohol. The red, green, and blue lines correspond to measurements obtained using the same buffer with ethanol, 1-propanol, and 2-propanol, respectively, at 4%. (**b**) Schematic representation of the long-axis length of a single DNA molecule. (**c**) Fluorescence miscroscopic image of a single T4GT7 DNA.

**Figure 4 polymers-15-00149-f004:**
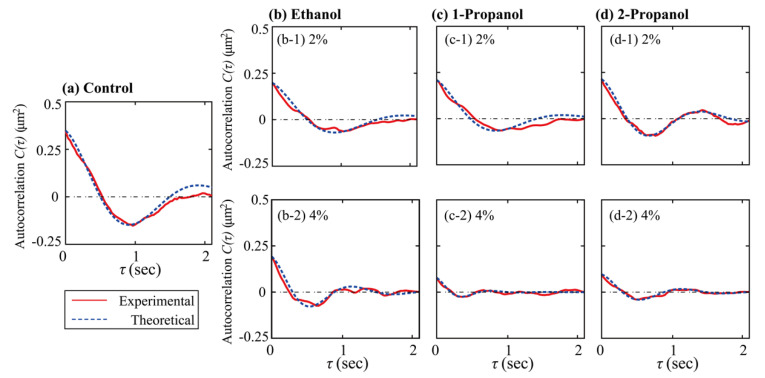
Autocorrelation of the time-dependent fluctuation of the long-axis length of single T4 GT7 DNA molecules (166 kbp). The curve was fitted based on Equation (2). (**a**) Fluctuations measured in the Tris-HCl buffer solution without alcohol (control), (**b**) with ethanol, (**c**) with 1-propanol, or (**d**) with 2-propanol.

**Figure 5 polymers-15-00149-f005:**
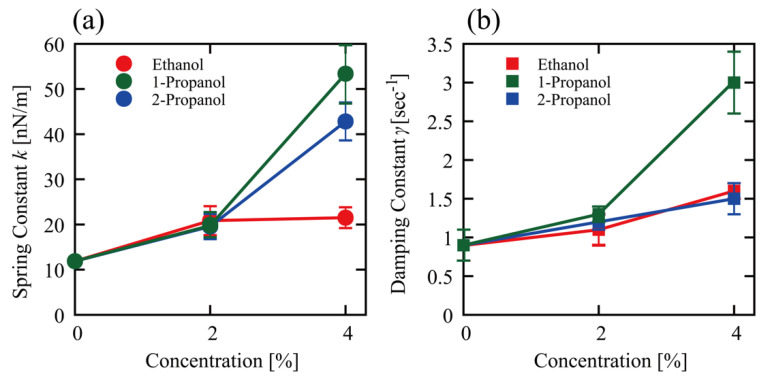
Viscoelasticity of single T4 GT7 DNA molecules (166 kbp) at different concentrations of ethanol, 1-propanol, and 2-propanol, as evaluated from the autocorrelation function shown in Figure 4. (**a**) Spring constant *k* and (**b**) damping constant *γ*. Data presented as mean ± SEM.

## Data Availability

All data presented in this study are contained within the article and Appendix A.

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
