# Peer review of "Activation/Inhibition of Gene Expression Caused by Alcohols: Relationship with the Viscoelastic Property of a DNA Molecule"

_polymers, 2022, doi:10.3390/polym15010149_

Round 1
Reviewer 1 Report
In this study, authors concluded that the alcohol isomers on gene expression correlate with the changes in viscoelastic mechanical property of DNA molecules. However, there is no gene expression data in this manuscript. Authors have to address this point either by displaying the gene expression-related data or re-writing some parts to justify this conclusion.
Author Response
Comments and Suggestions for Authors:
In this study, authors concluded that the alcohol isomers on gene expression correlate with the changes in viscoelastic mechanical property of DNA molecules. However, there is no gene expression data in this manuscript. Authors have to address this point either by displaying the gene expression-related data or re-writing some parts to justify this conclusion.
Response (with yellow back-color):
In response to the comment by the Reviewer, in the revised version we have added the detailed experimental data on the gene expression as Table A1 in the Appendix with the title of “Detailed gene expression data on the efficiency of (a) gene expression (TX-TL) and (b) translation (TL) at different alcohol concentrations of ethanol, 1-propanol and 2-propanol”.
In addition, we have refined the sentence in the Abstract (on lines 12-15 in the revised version) to reveal our experimental effort concerning the effect of alcohols on gene expression, as follows:
Revision:
In the present report, we carried out experiments to make clear how alcohols affect the efficiency of transcription-translation (TX-TL) and translation (TL) by adapting cell-free gene expression systems with plasmid DNA and RNA templates, respectively.
Reviewer 2 Report
The manuscript "Activation/Inhibition of Gene Expression Caused by Alcohols: Relationship with the Viscoelastic Property of a DNA Molecule" by Fujino et al. observed the behavior of low concentration alcohols on the TX/TL behavior of DNA, and connected with the viscoelastic property of DNA in the corresponding environment. The connection is clear. I'll recommend the publication of the manuscript with the following minor points made clear:
1. In line 61, the authors mentioned ""effects among low molecular-weight alcohols needs to be done." I would like to see a clearer motivation on why low molecular weight is important. Either it's less studied, or more commonly used, or some other brief reasons for this.
2. The concentration focused in the manuscript is below 10%. The author also mentioned the effect of alcohol intake. I'd like to see a brief comment on, what level of intake would result in such concentration. Or, more generally, under what situation would one encounter such concentration, especially in a TX/TL environment.
3. In line 175, the author utilized the autocorrelation of long-axis length to extract the viscoelastic property of DNA molecules. Does the autocorrelation function itself has physical meanings?
4. In line 336, the author mentioned the the solution conditions for gene expression and viscoelasticity measurement is different, so the alcohol concentration cannot be directly compared. Since the author is using one experiment to explain the other, they need to be comparable. So I would like to see a brief comment on this, saying how can concentration in each condition can be regarded as similar with each other, and which properties/parameters are deterministic in such comparison.
Reviewer 3 Report
The presented study is interesting and original in itself, since it is clearly shown that a small amount of alcohol (2-3%) increases gene expression by 4-5 times. In addition, there is evidence that the activating effects of alcohol isomers on gene expression correlate with their effect on the viscoelastic properties of DNA molecules. In general, I liked the reviewed text. The manuscript is well focused and properly organized, the experiment is clear, and the results are interesting and accurate.
Author Response
Comments and Suggestions for Authors
The presented study is interesting and original in itself, since it is clearly shown that a small amount of alcohol (2-3%) increases gene expression by 4-5 times. In addition, there is evidence that the activating effects of alcohol isomers on gene expression correlate with their effect on the viscoelastic properties of DNA molecules. In general, I liked the reviewed text. The manuscript is well focused and properly organized, the experiment is clear, and the results are interesting and accurate.
Reply by the authors:
We greatly thank for the positive comments by the Reviewer.
Round 2
Reviewer 1 Report
No more revision is needed.